# Early Identification of Sepsis-Induced Acute Kidney Injury by Using Monocyte Distribution Width, Red-Blood-Cell Distribution, and Neutrophil-to-Lymphocyte Ratio

**DOI:** 10.3390/diagnostics14090918

**Published:** 2024-04-28

**Authors:** Yi-Hsiang Pan, Hung-Wei Tsai, Hui-An Lin, Ching-Yi Chen, Chun-Chieh Chao, Sheng-Feng Lin, Sen-Kuang Hou

**Affiliations:** 1Department of Emergency Medicine, Taipei Medical University Hospital, Taipei 110, Taiwan; b101100078@tmu.edu.tw (Y.-H.P.); kevin575168@gmail.com (H.-W.T.); sevenoking219@gmail.com (H.-A.L.); chaosees@gmail.com (C.-C.C.); 2Graduate Institute of Injury Prevention and Control, College of Public Health, Taipei Medical University, Taipei 110, Taiwan; 3Department of Emergency Medicine, School of Medicine, College of Medicine, Taipei Medical University, Taipei 110, Taiwan; 4Department of Internal Medicine, School of Medicine, College of Medicine, Taipei Medical University, Taipei 110, Taiwan; 133007@h.tmu.edu.tw; 5Division of Nephrology, Department of Internal Medicine, Taipei Medical University Hospital, Taipei 110, Taiwan; 6School of Public Health, College of Public Health, Taipei Medical University, 250 Wu-Hsing Street, Taipei 110, Taiwan; 7Department of Public Health, School of Medicine, College of Medicine, Taipei Medical University, Taipei 110, Taiwan; 8Center of Evidence-Based Medicine, Taipei Medical University Hospital, Taipei 110, Taiwan

**Keywords:** critical care, diagnostic tests, sepsis, renal, acute kidney injury

## Abstract

Sepsis-induced acute kidney injury (AKI) is a common complication in patients with severe illness and leads to increased risks of mortality and chronic kidney disease. We investigated the association between monocyte distribution width (MDW), red-blood-cell volume distribution width (RDW), neutrophil-to-lymphocyte ratio (NLR), sepsis-related organ-failure assessment (SOFA) score, mean arterial pressure (MAP), and other risk factors and sepsis-induced AKI in patients presenting to the emergency department (ED). This retrospective study, spanning 1 January 2020, to 30 November 2020, was conducted at a university-affiliated teaching hospital. Patients meeting the Sepsis-2 consensus criteria upon presentation to our ED were categorized into sepsis-induced AKI and non-AKI groups. Clinical parameters (i.e., initial SOFA score and MAP) and laboratory markers (i.e., MDW, RDW, and NLR) were measured upon ED admission. A logistic regression model was developed, with sepsis-induced AKI as the dependent variable and laboratory parameters as independent variables. Three multivariable logistic regression models were constructed. In Model 1, MDW, initial SOFA score, and MAP exhibited significant associations with sepsis-induced AKI (area under the curve [AUC]: 0.728, 95% confidence interval [CI]: 0.668–0.789). In Model 2, RDW, initial SOFA score, and MAP were significantly correlated with sepsis-induced AKI (AUC: 0.712, 95% CI: 0.651–0.774). In Model 3, NLR, initial SOFA score, and MAP were significantly correlated with sepsis-induced AKI (AUC: 0.719, 95% CI: 0.658–0.780). Our novel models, integrating MDW, RDW, and NLR with initial SOFA score and MAP, can assist with the identification of sepsis-induced AKI among patients with sepsis presenting to the ED.

## 1. Introduction

Sepsis, a life-threatening systemic reaction to infection, poses a grave threat because it leads to the dysfunction of multiple organs that often results from an uncontrolled host response. It is a prominent cause of mortality among critically ill patients in hospitals that leads to decreased quality of life. Since its initial consensus definition was established in 1991, sepsis has remained a leading cause of morbidity and mortality globally. Sepsis can cause various complications, including brain dysfunction [1], cardiomyopathy [2], hepatic dysfunction [3], coagulopathy [4], and acute kidney injury (AKI) [5]. AKI is a particularly well-known complication; approximately 50% of AKI cases are caused by sepsis [6]. AKI increases mortality rates and the risk of progression to chronic kidney disease (CKD) in patients with sepsis [7]. The pathophysiology of AKI in sepsis is complex and multifactorial, involving intrarenal hemodynamic changes, endothelial dysfunction, inflammatory cell infiltration in the renal parenchyma, intraglomerular thrombosis, and tubular obstruction due to necrotic cells and debris [8]. Various risk factors for sepsis-induced AKI have been identified, including hypovolemia, diabetes mellitus, and exposure to nephrotoxic agents, such as contrast media, angiotensin-converting enzyme inhibitors, and metformin [9,10,11,12,13].

Early identification and intervention in sepsis-induced AKI are pivotal in improving disease prognosis and facilitating patient recovery. Many biomarkers and scoring systems for predicting sepsis-induced AKI have been researched. These biomarkers, including neutrophil gelatinase-associated lipocalin, cystatin C, kidney-injury molecule-1 (KIM-1), interleukin 18 (IL-18), urinary insulin-like growth factor-binding protein-7 (IGFBP-7), urinary tissue inhibitor of metalloproteinase 2 (TIMP-2), calprotectin, urine angiotensinogen, and liver fatty acid binding protein [14], are not readily available in most emergency department (ED) settings. In addition, measurement and interpretation of other physiological factors related to AKI, such as the renal resistive index (RRI) [15,16] and central venous pressure (CVP) [15], can only be completed by more experienced physicians. The diagnostic accuracy of the RRI, which is calculated through ultrasound, may vary with the level of operator proficiency, and CVP, an invasive hemodynamic parameter necessitating central venous catheter placement, is typically determined only for patients whose condition is deteriorating to septic shock. The processes for recording the RRI and CVP are complex and require certified specialists. Conversely, markers related to complete blood count (CBC), such as the red-blood-cell distribution width (RDW) [17] and neutrophil-to-lymphocyte ratio(NLR) [18,19], are more readily available, and their association with AKI has been extensively studied. Furthermore, hemodynamic parameters and scoring systems, such as the sequential organ-failure assessment (SOFA) score, the cardiovascular SOFA score [20], mean arterial pressure (MAP) [20], and the acute physical and chronic health evaluation (APACHE) II score [21] have been explored in the literature.

Red-blood-cell distribution width (RDW), a component of the CBC, represents the broadness of erythrocyte size distribution. This widely available erythrocyte index can be easily obtained through routine blood tests conducted in EDs. RDW has conventionally been used for the differentiation and classification of various types of anemia [22]. However, in the past decade, numerous studies have highlighted RDW as a valuable biomarker for predicting many clinical conditions, including cardiovascular disease [23], respiratory disease [24], ischemic stroke [25], cancer [26], hepatitis B [27], and sepsis [28], Additionally, research has revealed an association between RDW and AKI among patients with sepsis after cardiac surgery and in coronary care units [29]

The NLR, derived from a standard CBC, is calculated as the ratio of neutrophils to lymphocytes. Numerous studies have validated its applicability as a prognostic marker in various inflammation-associated and physiological stress-induced diseases, including sepsis, pneumonia, cancer, cardiovascular diseases, and kidney diseases [30,31,32]. Over the past 5 years, many studies have explored the association between the NLR and sepsis-induced AKI [33,34,35].

Monocyte activation is a hallmark of early inflammatory response and sepsis [36]. Monocyte distribution width (MDW), a parameter used to assess morphological changes in blood monocytes through the application of automated hemocyte analyzers, was reported to increase during the early stage of sepsis and to have diagnostic and predictive capabilities [37]. MDW information is readily available and part of routine laboratory testing included in the CBC; it can easily be obtained in the ED. Although several studies have highlighted MDW’s potential for early sepsis prediction, studies investigating the association between MDW and sepsis-induced AKI are few. The current study investigated the associations between several parameters, namely MDW, RDW, the NLR, SOFA score, MAP, other risk factors, and sepsis-induced AKI in the ED.

## 2. Materials and Methods

### 2.1. Study Design and Data Collection

This retrospective study employed data from a prospective registry in the ED of Taipei Medical University Hospital. Taipei Medical University Hospital is a tertiary referral and academic hospital in Taipei, Taiwan with 750 beds. The study included patients with infectious diseases who presented to our ED from 1 January 2020 onward. Data on each patient’s age, sex, vital signs, Glasgow coma scale score, laboratory tests, and history of medical comorbidities (including hypertension, diabetes mellitus, coronary artery disease, cerebrovascular disease, end-stage renal disease, pulmonary disease, liver cirrhosis, and malignant disease); medication use in the past 3 months; and admission in the past 3 months were collected. Our study was approved by the Joint Institutional Review Board of Taipei Medical University (approval number: N201904066), which waived the requirement for informed consent because of the study’s use of anonymous and deidentified data. This study involved no patient or the public.

### 2.2. Participants

Patients eligible for inclusion were those registered in the prospective registry system of the ED between 1 January 2020, and 30 November 2020. Patients presenting to our ED who met the Sepsis-2 [38] criteria (as confirmed by the authors H-W Tsai and Y-H Pan), were examined by ED physicians and had completed laboratory tests within 2 h of arrival were included. Patients who were aged <20 years, did not have a definite diagnosis of infectious disease (according to a data quality assessment conducted by H-W Tsai and Y-H Pan), did not undergo laboratory tests conducted using a hematology analyzer at the ED, received regular hemodialysis, and lacked baseline creatinine data for AKI diagnosis (according to the Kidney Disease Improving Global Outcomes [KIDGO] guidelines) were excluded [39].

### 2.3. Outcome Measures

The primary outcome of this study was a diagnosis of AKI in accordance with the KDIGO guidelines [39]. Patients with sepsis were identified on the basis of the Sepsis-2 consensus criteria, which included documented infection and fulfillment of at least two of the following systemic inflammatory response syndrome (SIRS) criteria: (1) a body temperature of <36 °C or >38 °C, (2) heart rate of >90 beats/min, (3) respiratory rate of >20 breaths/min or partial CO_2_ pressure of <32 mmHg, and (4) WBC count of <4000 or >12,000 cells/mm^3^ (or with 10% bands) [38]. Patients with AKI were defined as those with a creatinine elevation of at least 0.3 mg/dL or 1.5 times baseline levels [40]. On the basis of these criteria, the included patients were categorized into sepsis-induced AKI and non-AKI groups.

### 2.4. Biomarker Measurement

Since 1 January 2020, MDW has been used to assess infectious diseases at Taipei Medical University Hospital. MDW is analyzed using a Beckman Coulter UniCel DxH 900 analyzer (Beckman Coulter Taiwan, Taiwan Branch). It is computed as the standard deviation of a series of monocyte cell volume measurements. Our study also incorporated the NLR and RDW as biomarkers for sepsis-induced AKI.

### 2.5. Statistical Analysis

The Shapiro–Wilk test was used to determine whether continuous variables, including age, MDW, RDW, initial SOFA score, MAP, and the NLR, adhered to the parametric assumption for normality. In cases where the Shapiro–Wilk tests indicated a violation of this assumption, the Mann–Whitney U test was used to compare groups with and without sepsis-induced AKI. Categorical variables were analyzed using Pearson’s chi-squared test (or Fisher’s exact test). Continuous variables are presented as medians (interquartile ranges), whereas categorical variables are expressed as proportions. Univariable and multivariable logistic regression models were employed to obtain odds ratios (ORs) and corresponding 95% confidence intervals (CIs) to assess the associations between sepsis-induced AKI and other variables. The receiver operating-characteristic (ROC) curve and Youden’s index were used to determine the optimal cutoff (the point with the maximum value of sensitivity + specificity −1). Significant predictors (*p* < 0.05) from the univariate analysis were included in the multivariable logistic regression, and the ROC curve was used to identify the most predictive combination of these variables. Model selection was conducted through backward elimination with the lowest Akaike information criterion. A *p* value of <0.05 was considered significant, and all statistical analyses were performed using IBM SPSS Statistics 18 (SPSS, Chicago, IL, USA).

## 3. Results

### 3.1. Participant Characteristics

A total of 19,792 patients presented to our ED between January 2020 and November 2020. After a meticulous assessment of the data quality, 8698 patients were confirmed to have an infectious disease. Sepsis was confirmed on the basis of the Sepsis-2 consensus criteria in 308 of these patients. Patients with creatinine levels incongruent with the diagnostic criteria for AKI and those undergoing hemodialysis were excluded. Ultimately, 271 patients were included in the study, as illustrated in the flowchart presented in Figure 1. These 271 patients were categorized into sepsis-induced AKI (112, 41%) and non-AKI (159, 59%) groups by referencing the KDIGO guidelines. The demographic characteristics of the enrolled patients are presented in Table 1. Among the patients, 120 were men and 151 were women, and the mean age was 70.04 ± 16.95 years. Appendix A presents the results of the Shapiro–Wilk tests. Significant differences were observed between the groups without and with sepsis-induced AKI in terms of MDW (median of 21.80 vs. 26.20, *p* < 0.001), initial SOFA score (median of 2.00 vs. 3.00, *p* = 0.008), and MAP (median of 97.83 vs. 86.00, *p* < 0.001).

### 3.2. Univariate Logistic Regression

Table 2 presents the results of univariate logistic regression analysis of the variables. MDW (OR: 1.032, 95% CI: 1.012–1.061, *p* = 0.003), RDW (OR: 1.115, 95% CI: 1.025–1.212, *p* = 0.011), initial SOFA score (OR: 1.283, 95% CI: 1.143–1.442, *p* < 0.001), MAP (OR: 0.975, 95% CI: 0.963–0.987, *p* < 0.001), and the NLR (OR: 1.014, 95% CI: 1.001–1.027, *p* = 0.029) exhibited significant associations with sepsis-induced AKI.

### 3.3. Multivariate Logistic Regression

We identified significant variables from the univariable analysis for inclusion in the multivariable logistic regression. Three models were established, with each incorporating different laboratory parameters alongside the initial SOFA score and MAP. Table 3 presents the results for the three multivariable logistic regression models. In Model 1, MDW (OR: 1.025, 95% CI: 1.001–1.050, *p* = 0.044), initial SOFA score (OR: 1.202, 95% CI: 1.061–1.361, *p* = 0.004), and MAP (OR: 0.982, 95% CI: 0.970–0.995, *p* = 0.008) demonstrated significant associations with sepsis-induced AKI. In Model 2, RDW (OR: 1.070, 95% CI: 0980–1.167, *p* = 0.130), initial SOFA score (OR: 1.204, 95% CI: 1.063–1.364, *p* = 0.004), and MAP (OR: 0.982, 95% CI: 0.969–0.994, *p* = 0.005) were significantly associated with sepsis-induced AKI. In Model 3, the NLR (OR: 1.010, 95% CI: 0.997–1.024, *p* = 0.116), initial SOFA score (OR: 1.212, 95% CI: 1.070–1.372, *p* = 0.002), and MAP (OR: 0.982, 95% CI: 0.969–0.995, *p* = 0.006) were significantly associated with sepsis-induced AKI.

### 3.4. ROC Curve and Area under the Curve

Figure 2 presents ROC curves depicting different combinations of variables and their corresponding areas under the curve (AUCs). The AUCs of MDW, RDW, initial SOFA score, MAP, and the NLR were 0.652 (95% CI: 0.528–0.721), 0.590 (95% CI: 0.522–0.659), 0.661 (95% CI: 0.596–0.726), 0.669 (95% CI: 0.604–0.734), and 0.573 (95% CI: 0.503–0.643), respectively, for predicting sepsis-induced AKI. Of the three models, Model 1 (MDW, initial SOFA score, and MAP) demonstrated the highest AUC (0.728; 95% CI: 0.668–0.789). Notably, the combination of MDW with the initial SOFA score and MAP exhibited the highest predictive value (AUC = 0.728) for sepsis-induced AKI. Additional biomarkers were incorporated into Model 2 (RDW, initial SOFA score, and MAP with an AUC of 0.712 [95% CI: 0.651–0.774]) and Model 3 (the NLR, initial SOFA score, and MAP with an AUC of 0.719 [95% CI: 0.658–0.780]).

## 4. Discussion

In this study, approximately 40% of the patients experienced sepsis-related AKI, which is a prevalence consistent with previous findings. AKI can substantially influence the likelihood of subsequent mortality, with estimates indicating it leads to a 6- to 8-fold increase in likelihood [41]. However, half of AKI cases manifest covertly before patients present to the ED [6,42,43]. Timely and accurate detection of AKI emergence is imperative in the ED setting. Physicians must have practical tools that can improve the diagnosis of emergent AKI, such as biomarkers that can be assessed in ED settings.

To date, no study has directly investigated the cellular and molecular mechanisms underlying the association between MDW and sepsis-induced AKI. In the current study, we proposed the following as a potential mechanism: upon microbial invasion, the body initiates a hyperactive dysregulated innate immune response. This response triggers the release of a cascade of proinflammatory molecules, which activate the complement system and cellular innate immunity, ultimately contributing to the development of sepsis-induced AKI [44].

Monocytes, commonly observed bone-marrow-derived mononuclear cells in peripheral blood, constitute 5–10% of circulating immune cells [45]. They play a pivotal role in the innate immune system and are crucial during the initial phase of inflammation and tissue remodeling [46,47]. We hypothesize that assessing the dynamic morphological changes of monocytes through the measurement of MDW would reveal an association between MDW and sepsis-induced AKI.

Several studies have highlighted the critical roles of ischemia–reperfusion, tissue remodeling, and inflammation in the pathophysiology of AKI [48,49,50]. Ischemia or reperfusion events in the kidney lead to dysfunction of renal tubular epithelial cells, vascular endothelial cells, and leukocytes. This disruption in kidney immune homeostasis contributes to inflammation, ultimately leading to cell death in kidney parenchymal cells and culminating in AKI [51,52].

In our three models, the combination of MDW with the initial SOFA score and MAP had the highest predictive value (AUC = 0.728) for sepsis-induced AKI. These models can be practically applied in the ED because the biomarkers and vital-sign parameters can be easily assessed. Several studies have investigated the association between MDW and sepsis [37,53,54], and their findings have indicated that MDW exhibits optimal diagnostic accuracy for sepsis. A study reported that, at an MDW cutoff of 23.5 U, the AUC reached 0.964 [37]. Other studies have indicated that, when qSOFA > 1, the NLR > 9, the PLR > 210, and MDW > 20, the AUC reached 0.757, enabling early sepsis prediction in ED patients [54]. In our study, MDW exhibited an AUC of 0.652. Notably, MWD had the highest predictive effect when the cutoff was set at 25 U, with a sensitivity of 83.9% and specificity of 16.98%. In addition, because the initial SOFA score and MAP have higher specificity, including these parameters in assessments can offset the lower specificity of MDW and thereby enhance the predictive value of our model. Furthermore, MDW exhibited a higher AUC than RDW and the NLR (0.652 vs. 0.590 vs. 0.573), indicating it is superior with respect to predicting AKI in patients with sepsis. Therefore, MDW can be considered a viable alternative to RDW and the NLR, as was also reported in previous studies [17,18] for AKI prediction in patients with sepsis.

RDW and NLR are cost-effective biochemical parameters that can easily be obtained in the ED. Several studies have investigated their associations with AKI [29,35,55]. Although the mechanisms underlying the association between RDW and sepsis-induced AKI remain elusive, two studies have proposed plausible explanations. One study suggested that inflammation processes inhibit iron metabolism, bone-marrow function, and the proliferation and maturation of erythrocytes, leading to an increase in RDW values [56]. Furthermore, the multiorgan dysfunction observed in sepsis cases often involves severe oxidative stress. A study highlighted the pivotal role of oxidative stress in the cell cycle, differentiation, and maturation of erythroid cells and the association between RDW and several oxidative stress biomarkers [57]. The NLR is a biomarker that involves two aspects of the immune system: the innate response, which is predominantly mediated by neutrophils, and the adaptive response, which is primarily attributable to lymphocytes [58]. Generally, inflammatory diseases are associated with an increase in the neutrophil count and a decrease in the lymphocyte count as they progress. Several studies have employed the NLR to predict mortality and morbidity [59,60]. The NLR was reported to be more reliable than neutrophil or lymphocyte counts alone in predicting sepsis outcomes [59]. Studies investigating the association between AKI and the NLR have indicated that the NLR may be a predictor of AKI in patients who have undergone surgical or radiological procedures [55,61,62]. Moreover, a study reported a potential association between the NLR and AKI in patients with sepsis [35].

The AUC of RDW for long-term prognosis in critically ill patients with AKI was reported to be 0.718 [17]. In a study involving hospitalized patients with AKI, increased mortality rates were observed in the high RDW group (RDW > 13.95, AUC = 0.63) and the high NLR group (NLR > 5.51, AUC = 0.65) [63]. Additionally, these patients had a higher risk of requiring renal replacement therapy [63]. A retrospective study demonstrated that in patients with sepsis and septic shock, the AUC of the NLR for predicting septic AKI was 0.656 [35]. Another study used 16 independent predictors for sepsis-induced AKI in critically ill patients and indicated that they had excellent predictive ability (AUC = 0.857) [64]. However, several key variables indicating AKI risk in these studies are comorbidities, and information on these variables may be lacking or missing due to patients’ expressions or inconsistent clinical documentation. Additionally, collecting more blood samples to analyze the aforementioned laboratory parameters can be time-consuming and impose additional clinical burdens. Our approach involves the use of only three parameters for risk evaluation. Our findings revealed that the AUCs of RDW and the NLR were only 0.590 and 0.573, respectively. Model 1 demonstrated the highest predictive value when the RDW was combined with the initial SOFA score and MAP. Nevertheless, Model 2 and Model 3 also exhibited favorable predictive values. The combination of RDW, initial SOFA, and MAP resulted in an AUC of 0.712, whereas that of the NLR, initial SOFA, and MAP resulted in an AUC of 0.719. In real-world settings, however, not all hospitals are able to measure MDW. In such settings, physicians could use the RDW or the NLR models for evaluation. The establishment of a diagnostic model for sepsis and AKI can facilitate early diagnosis and enable the provision of effective treatments for critically ill patients. Notably, the biomarker and vital-sign parameters used in our models can be easily obtained in the ED.

Although studies have identified numerous biomarkers for the prediction of sepsis-induced AKI, some require time-consuming testing. In the ED, most biomarker parameters and test results (such as MDW, RDW, the NLR, and MAP) can be obtained within an hour, facilitating early recognition of sepsis-induced AKI. Additionally, the initial SOFA score can usually be obtained within 90 min. The current study combined several parameters and biomarkers, which can enable the identification of the risk of sepsis-induced AKI and the initiation of associated evaluations and treatments within 90 min of arrival at the ED. Early awareness of sepsis-induced AKI can expedite decision-making. 

In this study, we identified biomarkers and parameters that are readily available, are cost-effective, and require less time and effort to obtain. We developed novel models with high accuracy in terms of AKI prediction that can easily be employed in various medical institutions and settings. However, this study has several limitations that should be acknowledged. First, because this was a single-center retrospective study, residual confounding cannot be entirely ruled out. Second, although several risk factors for AKI were included in the multivariable analysis, some potential risk factors were not explored. Third, the definition of sepsis has evolved over time, with the most recent consensus definition being the Sepsis-3 definition established in 2016. Variations in diagnostic standards for sepsis may result in different study groups for sepsis-induced AKI. We used the Sepsis-2 consensus criteria in our study because they have higher sensitivity in detecting sepsis, whereas the Sepsis-3 criteria have favorable performance in predicting mortality [65]. Additionally, the precise and timely measurement of urine-output status in the ED can be challenging, and therefore, we defined AKI on the basis of creatinine levels, which are less sensitive to AKI detection than urine-output status. Fourth, the majority of the enrolled patients were of Han Chinese ethnicity. Therefore, additional studies are required to validate the optimal thresholds of MDW, RDW, and the NLR for other ethnic groups. Fifth, our study focused specifically on the ED, and long-term outcome evaluations for the included patients were not conducted. Finally, we did not conduct serial testing of MDW, RDW, and the NLR, and therefore, their correlation with the trend of decreasing renal function remains to be investigated. Additional studies are required to establish a more comprehensive understanding of the association between sepsis and AKI. Nevertheless, because of our use of strict definitions for sepsis and the inclusion of multiple variables in our data analysis, this study provides valuable information that can serve as a reference for future investigations.

Our study proposed a model using MDW in conjunction with the initial SOFA score and MAP to detect AKI in patients with sepsis presenting to the ED. However, additional multicenter studies with larger sample sizes are warranted to validate and refine this model. Lastly, we provided the summary at a glance as follows.

Sepsis-induced acute kidney injury (AKI) increases morbidity and mortality.An increase in blood cell anisocytosis is associated with sepsis-induced AKI.MDW is a novel biomarker for predicting sepsis-induced AKI.

## Figures and Tables

**Figure 1 diagnostics-14-00918-f001:**
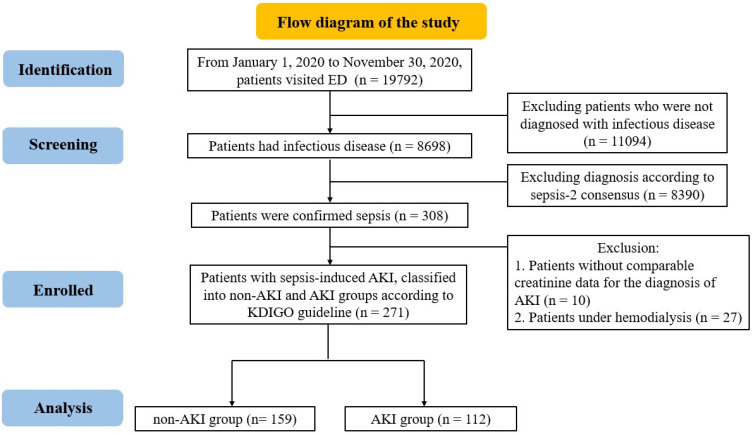
Flowchart of participant recruitment.

**Figure 2 diagnostics-14-00918-f002:**
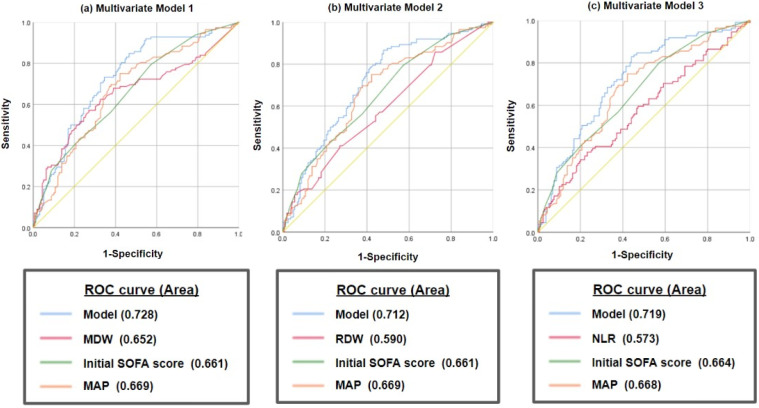
Receiver operating-characteristic curves and area under the curve of multivariate models. The yellow line is the diagonal reference line.

**Table 1 diagnostics-14-00918-t001:** Patient characteristics (*N* = 271).

Variables	Non-AKI	AKI	*p* Value
	(*N* = 159)	(*N* = 112)	
Age (years)	73.00 (22.00)	72.00 (23.00)	0.322
Age subgroups *N*, (%)			0.907
20–45 years	14 (8.8%)	12 (10.7%)	
46–70 years	54 (34.0%)	40 (35.7%)	
71–95 years	86 (54.1%)	56 (50.0%)	
96–120 years	5 (3.1%)	4 (3.6%)	
Sex, *N* (%)			0.883
Female	88 (55.3%)	63 (56.3%)	
Male	71 (44.7%)	49 (43.8%)	
DM	53 (33.3%)	42 (37.5%)	0.479
CKD	141 (88.7%)	102 (91.1%)	0.524
Recent antibiotic use	58 (36.5%)	38 (33.9%)	0.666
Recent contrast	28 (17.6%)	17 (15.2%)	0.596
Recent NSAIDs use	10 (6.3%)	8 (7.1%)	0.781
Recent ARB use	41 (25.8%)	39 (34.8%)	0.108
Recent beta-blocker use	51 (32.1%)	35 (31.3%)	0.886
Recent anti-PLT use	41 (25.8%)	27 (24.1%)	0.754
Recent diuretics use	34 (21.4%)	24 (21.4%)	0.993
MDW	21.80 (7.00)	26.20 (13.00)	<0.001 *
RDW	14.00 (2.90)	15.00 (2.90)	0.136
initial SOFA	2.00 (2.00)	3.00 (3.00)	0.008 *
MAP	97.83 (26.20)	86.00 (21.00)	<0.001 *
NLR	12.65 (13.23)	14.55 (19.82)	0.180

Continuous variables are expressed as median (interquartile range). Abbreviations: AKI, acute kidney injury; ARB, angiotensin-receptor blockers; CKD, chronic kidney disease; DM, diabetes mellitus; MAP, mean arterial pressure; MDW, monocyte distribution width; N, number; NSAID, nonsteroidal anti-inflammatory drug; NLR, neutrophil-to-lymphocyte ratio; PLT, platelet; RDW, red-blood-cell volume distribution width; SOFA, sepsis-related organ-failure assessment. * Statistical significance was defined as *p* < 0.05.

**Table 2 diagnostics-14-00918-t002:** Simple logistic regression analyses for predicting AKI in patients with sepsis presenting to the emergency department.

Variables	OR (95% CI)	*p* Value
MDW	1.032 (1.012–1.061)	0.003
RDW	1.115 (1.025–1.212)	0.011
initial SOFA	1.283 (1.143–1.442)	<0.001
MAP	0.975 (0.963–0.987)	<0.001
NLR	1.014 (1.001–1.027)	0.029
Age	0.994 (0.980–1.009)	0.427
Gender	1.037 (0.637–1.688)	0.883
DM	0.833 (0.503–1.381)	0.479
CKD	0.768 (0.340–1.733)	0.525
Recent antibiotic use	0.894 (0.538–1.485)	0.666
Recent contrast	0.837 (0.434–1.617)	0.597
Recent NSAIDs use	1.146 (0.438–3.002)	0.781
Recent ARB use	0.650 (0.384–1.101)	0.109
Recent beta-blocker use	1.039 (0.618–1.748)	0.886
Recent anti-PLT use	1.094 (0.625–1.915)	0.754
Recent diuretics use	0.997 (0.553–1.798)	0.993

Abbreviations: MDW, monocyte distribution width; RDW, red-blood-cell distribution width; SOFA, sepsis-related organ-failure assessment; MAP, mean arterial pressure; NLR, neutrophil-to-lymphocyte ratio; DM, diabetes mellitus; CKD, chronic kidney disease; NSAID, nonsteroidal anti-inflammatory drug; ARB, angiotensin-receptor blockers; PLT, platelet; OR, odds ratio; CI, confidence interval.

**Table 3 diagnostics-14-00918-t003:** Multivariate logistic regression analyses for predicting AKI in patients with sepsis presenting to the emergency department.

	Multivariate Analysis (Model 1)		Multivariate Analysis (Model 2)		Multivariate Analysis (Model 3)	
Variables	OR (95% CI)	*p* value	OR (95% CI)	*p* value	OR (95% CI)	*p* value
MDW	1.025 (1.001–1.050)	0.044				
RDW			1.070 (0.980–1.167)	0.130		
initial SOFA	1.202 (1.061–1.361)	0.004	1.204 (1.063–1.364)	0.004	1.212 (1.070–1.372)	0.002
MAP	0.982 (0.970–0.995)	0.008	0.982 (0.969–0.994)	0.005	0.982 (0.969–0.995)	0.006
NLR					1.010 (0.997–1.024)	0.116

Abbreviations: MDW, monocyte distribution width; RDW, red-blood-cell volume distribution width; SOFA, sepsis-related organ-failure assessment; MAP, mean arterial pressure; NLR, neutrophil-to-lymphocyte ratio; OR, odds ratio; CI, confidence interval.

## Data Availability

Since the legal restrictions imposed by the government of Taiwan on the distribution of the personal health data in relation to the “Personal Information Protection Act”, requests for data need a formal proposal to the Joint of Institutional Review Board and the Office of Human Research of Taipei Medical University.

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
