# Peer review of "Early Identification of Sepsis-Induced Acute Kidney Injury by Using Monocyte Distribution Width, Red-Blood-Cell Distribution, and Neutrophil-to-Lymphocyte Ratio"

_diagnostics, 2024, doi:10.3390/diagnostics14090918_

Round 1

Reviewer 1 Report

Comments and Suggestions for Authors

Well-written paper. Congratulations to the authors.

Minor comment: The referencing (journal names ) abbreviation needs to be standardized.

Author Response

We thank the reviewers for the constructive comments. We have made revisions to the manuscript to address the questions and comments raised by the reviewers. We have highlighted changes made to the original version by setting the text color to red. Our specific responses to each comment are as follows:

Reviewer 1

Q: The referencing (journal names) abbreviation needs to be standardized.

A: Thank you for the comment. In our revised manuscript, we have used the required reference style (ACS citation style) for journal diagnostics. The order, author names, journal names and published years were revised accordingly. (Please see references section, the revised manuscript).

Reviewer 2 Report

Comments and Suggestions for Authors

The manuscript titled "Anisocytosis in Red Blood Kidney Cells and Monocytes is Associated with Sepsis-Induced Acute Injury" presents valuable insights that could prove to be highly informative.

Minor editing of English language required

- Limitations of the study should be added

In the discussion, it is important to try to explain the cellular and molecular mechanisms underlying the association between MDR and RDW with sepsis-induced AKI.

- What tests did you use to determine that your data were parametric? Add to materials and methods section

- Declaration section missing

- The flowchart in Figure 1 must comply with the specifications established by PRISMA.

- Why was the NLR not considered in the title? If it has greater power than the RDW

- A paragraph focused on NLR should be added to the discussion.

Comments on the Quality of English Language

Minor editing of English language required

Author Response

We thank the reviewers for the constructive comments. We have made revisions to the manuscript to address the questions and comments raised by the reviewers. We have highlighted changes made to the original version by setting the text color to red. Our specific responses to each comment are as follows:

Reviewer 2

Q: Minor editing of English language required

A: Thank you for the comment. In our revised manuscript, we have consulted the professional English editing service to check spelling, grammar, and readability of the texts. Please see the revised text in red color.

Q: Limitations of the study should be added

A: Thank you for the comment. In our revised manuscript, we have added six points of limitations in our current study design as follows. First, because this was a single-center retrospective study, residual confounding cannot be entirely ruled out. Second, although several risk factors for AKI were included in the multivariable analysis, some potential risk factors were not explored. Third, the definition of sepsis has evolved over time, with the most recent consensus definition being the Sepsis-3 definition established in 2016. Variations in diagnostic standards for sepsis may result in different study groups for sepsis-induced AKI. We used the Sepsis-2 consensus criteria in our study because they have higher sensitivity in detecting sepsis, whereas the Sepsis-3 criteria have favorable performance in predicting mortality. Additionally, precise and timely measurement of urine output status in the ED can be challenging, and therefore, we defined AKI on the basis of creatinine levels, which is less sensitive for AKI detection than urine output status is. Fourth, the majority of the enrolled patients were of Han Chinese ethnicity. Therefore, additional studies are required to validate the optimal thresholds of MDW, RDW, and the NLR for other ethnic groups. Fifth, our study focused specifically on the ED, and long-term outcome evaluations for the included patients were not conducted. Finally, we did not conduct serial testing of MDW, RDW, and the NLR, and therefore, their correlation with the trend of decreasing renal function remains to be investigated. Additional studies are required to establish a more comprehensive understanding of the association between sepsis and AKI. Nevertheless, because of our use of strict definitions for sepsis and inclusion of multiple variables in our data analysis, this study provides valuable information that can serve as a reference for future investigations. (Please see discussion section, page 16 and page 17)

Q: In the discussion, it is important to try to explain the cellular and molecular mechanisms underlying the association between MDW and RDW with sepsis-induced AKI.

A: Thank you for the comment. In our revised manuscript, we have addressed the possible mechanisms between MDW, RDW and sepsis-induced AKI as follows. (Please see discussion section)

To date, no study has directly investigated the cellular and molecular mechanisms underlying the association between MDW and sepsis-induced AKI. In the current study, we proposed the following as a potential mechanism: upon microbial invasion, the body initiates a hyperactive dysregulated innate immune response. This response triggers the release of a cascade of proinflammatory molecules, which activate the complement system and cellular innate immunity, ultimately contributing to the development of sepsis-induced AKI.

Monocytes, commonly observed bone marrow–derived mononuclear cells in peripheral blood, constitute 5%–10% of circulating immune cells. They play a pivotal role in the innate immune system and are crucial during the initial phase of inflammation and tissue remodeling. We hypothesize that assessing the dynamic morphological changes of monocytes through measurement of MDW would reveal an association between MDW and sepsis-induced AKI.

Several studies have highlighted the critical roles of ischemia‒reperfusion, tissue remodeling, and inflammation in the pathophysiology of AKI. Ischemia or reperfusion events in the kidney lead to dysfunction of renal tubular epithelial cells, vascular endothelial cells, and leukocytes. This disruption in kidney immune homeostasis contributes to inflammation, ultimately leading to cell death in kidney parenchymal cells and culminating in AKI. (Please see discussion section, page 12 and page 13)

Although the mechanisms underlying the association between RDW and sepsis-induced AKI remain elusive, two studies have proposed plausible explanations. One study suggested that inflammation processes inhibit iron metabolism, bone marrow function, and the proliferation and maturation of erythrocytes, leading to an increase in RDW values. Furthermore, the multiorgan dysfunction observed in sepsis cases often involves severe oxidative stress. A study highlighted the pivotal role of oxidative stress in the cell cycle, differentiation, and maturation of erythroid cells and the association between RDW and several oxidative stress biomarkers. (Please see discussion section, page 13 and page 14)

Q: What tests did you use to determine that your data were parametric? Add to materials and methods section

A: Thank you for the comment. First, we have performed the Shapiro-Wilk tests to examine whether our continuous variables, including age, MDW, RDW, initial SOFA score, MAP and NLR, fulfills the parametric assumption for normality. The Shapiro-Wilk tests, however, showed all of these continuous variables violated the assumption for normality. Accordingly, we have revised our analysis for continuous variables with the Mann-Whiteney U tests to replace the Student’s t tests. We have revised the methods and results sections with adding the above description. (Please see statistical analysis, page 9; Table 1; Supplemental table 1)

Q: Declaration section missing

A: Thank you for the comment. We added a paragraph of declaration section, funding, acknowledgements and author contribution before the reference list after discussion section according to the Journal’s requirement (referred to https://www.mdpi.com/journal/diagnostics/instructions#conflict). All authors declare no conflicts of interest. In addition, we have uploaded a formal declaration form with all authors’ signature.

Q: The flowchart in Figure 1 must comply with the specifications established by PRISMA.

A: Thank you for the comment. We have renewed Figure 1 according to the specifications established by PRISMA.

Q: Why was the NLR not considered in the title? If it has greater power than the RDW

A: Thank you for the comment. In our revised manuscript, we have replaced the original title with the revised one “Early identification of sepsis-induced acute kidney injury by using monocyte distribution width, red blood cell distribution, and neutrophil-to-lymphocyte ratio.”

Q: A paragraph focused on NLR should be added to the discussion.

A: Thank you for the comment. In our revised manuscript, we have added two paragraphs focused on the relationship between NLR and sepsis-induced AKI to the introduction and discussion section.

The NLR, derived from a standard CBC, is calculated as the ratio of neutrophils to lymphocytes. Numerous studies have validated its applicability as a prognostic marker in various inflammation-associated and physiological stress–induced diseases, including sepsis, pneumonia, cancer, cardiovascular diseases, and kidney diseases. Over the past 5 years, many studies have explored the association between the NLR and sepsis-induced AKI. (Please see introduction section, page 6)

he NLR is a biomarker that involves two aspects of the immune system: the innate response, which is predominantly mediated by neutrophils, and the adaptive response, which is primarily attributable to lymphocytes. Generally, inflammatory diseases are associated with an increase in neutrophil count and a decrease in lymphocyte count as they progress. Several studies have employed the NLR to predict mortality and morbidity. The NLR was reported to be more reliable than neutrophil or lymphocyte counts alone in predicting sepsis outcomes. Studies investigating the association between AKI and the NLR have indicated that the NLR may be a predictor of AKI in patients who have undergone surgical or radiological procedures. Moreover, a study reported a potential association between the NLR and AKI in patients with sepsis.

(Please see discussion section, page 14)

Reviewer 3 Report

Comments and Suggestions for Authors

Intriguing article with new methods useful for emergency department.

I have only a question for authors about the decision timing for this score in order to improve its usefulness for the ED. Adding further explanations in this field the article will acquire further fashion for readers and physicians.

Author Response

We thank the reviewers for the constructive comments. We have made revisions to the manuscript to address the questions and comments raised by the reviewers. We have highlighted changes made to the original version by setting the text color to red. Our specific responses to each comment are as follows:

Reviewer 3

Q:I have only a question for authors about the decision timing for this score in order to improve its usefulness for the ED. Adding further explanations in this field the article will acquire further fashion for readers and physicians.

A: Thank you for the comment. In our revised manuscript, we added a paragraph discussing about the decision timing for these scores in emergency department setting as follows. Although studies have identified numerous biomarkers for the prediction of sepsis-induced AKI, some require time-consuming testing. In the ED, most biomarker parameters and test results (such as MDW, RDW, the NLR, MAP) can be obtained within an hour, facilitating early recognition of sepsis-induced AKI. Additionally, the initial SOFA score can usually be obtained within 90 min. The current study combined several parameters and biomarkers, which can enable identification of the risk of sepsis-induced AKI and initiation of associated evaluations and treatments within 90 min of arrival at the ED. Early awareness of sepsis-induced AKI can expedite decision-making. (Please see discussion section, page 15 and page 16)

Round 2

Reviewer 2 Report

Comments and Suggestions for Authors

No additional comments

Comments on the Quality of English Language

Minimal editing of the English language is required and may be carried out in production.